# Barriers and Facilitators of Fruit and Vegetable Consumption in Renal Transplant Recipients, Family Members and Healthcare Professionals—A Focus Group Study

**DOI:** 10.3390/nu11102427

**Published:** 2019-10-11

**Authors:** Karin Boslooper-Meulenbelt, Olga Patijn, Marieke C. E. Battjes-Fries, Hinke Haisma, Gerda K. Pot, Gerjan J. Navis

**Affiliations:** 1Division of Nephrology, Department of Internal Medicine, University Medical Center Groningen, University of Groningen, Groningen 9700 RB, The Netherlands; g.j.navis@umcg.nl; 2Louis Bolk Insitute, Bunnik 3981 AJ, The Netherlands; o.patijn@louisbolk.nl (O.P.); g.pot@louisbolk.nl (G.K.P.); 3Population Research Center, Faculty Spatial Sciences, University of Groningen, Groningen 9747 AD, The Netherlands; h.h.haisma@rug.nl

**Keywords:** renal transplantation, nutrition, vegetable consumption, fruit consumption, barriers, focus groups

## Abstract

Low fruit and vegetable consumption is associated with poor outcomes after renal transplantation. Insufficient fruit and vegetable consumption is reported in the majority of renal transplant recipients (RTR). The aim of this study was to identify barriers and facilitators of fruit and vegetable consumption after renal transplantation and explore if certain barriers and facilitators were transplant-related. After purposive sampling, RTR (*n* = 19), their family members (*n* = 15) and healthcare professionals *(n* = 5) from a Dutch transplant center participated in seven focus group discussions (three each for RTR and family members, one with healthcare professionals). Transcripts were analyzed using social cognitive theory as conceptual framework and content analysis was used for identification of themes. Transplant-related barriers and facilitators were described separately. In categorizing barriers and facilitators, four transplant-related themes were identified: transition in diet (accompanied by, e.g., fear or difficulties with new routine), physical health (e.g., recovery of uremic symptoms), medication (e.g., cravings by prednisolone) and competing priorities after transplantation (e.g., social participation activities). Among the generic personal and environmental barriers and facilitators, food literacy and social support were most relevant. In conclusion, transplant-related and generic barriers and facilitators were identified for fruit and vegetable consumption in RTR. The barriers that accompany the dietary transition after renal transplantation may contribute to the generally poorer fruit and vegetable consumption of RTR. These findings can be used for the development of additional nutritional counseling strategies in renal transplant care.

## 1. Introduction

Renal transplantation is the preferred treatment for patients with end-stage renal disease (ESRD), with an improved quality of life and survival as compared with dialysis [1,2]. However, the life expectancy of renal transplant recipients (RTR) is still considerably lower than age-matched controls of the general population [3]. RTR often have poor cardio-metabolic health due to both conventional and transplant-specific risk factors, including lifestyle-related factors. Nutrition affects several cardiovascular risk factors after transplantation; healthy dietary patterns are associated with a lower risk of developing weight gain, metabolic syndrome or diabetes [4,5,6]. Therefore, improving dietary behavior is one of the important lifestyle measures that may contribute to better cardio-metabolic health outcomes of RTR.

Insufficient fruit and vegetable consumption is one of the dietary factors that is associated with poor outcomes after renal transplantation, including a higher risk of developing diabetes after transplantation (PTDM) and an increased cardiovascular mortality [7,8]. Unfortunately, in our transplant center we observed poor fruit and vegetable consumption in RTR, despite regular nutritional counseling by a renal dietician (outlined in Box 1) [7,8]. A recent study in 472 RTR showed that RTR have a lower median vegetable consumption than the general Dutch population (108 g/d versus 127 gr/d) [7,9]. It is not fully understood why RTR consume less fruit and vegetables than the general population. Insufficient fruit and vegetable consumption is often reported in dialysis patients and at least in part due to the dietary restrictions with limitation of the potassium load [10]. These dietary restrictions are no longer necessary in the majority of the RTR. Continuation of habitual potassium restrictions after transplantation could be a salient barrier for adequate fruit and vegetable consumption. This has been suggested previously as underlying cause of low fruit and vegetable consumption in RTR, which is accompanied by lower urinary potassium excretion [11,12].

In general, modifications in dietary behavior are known to be affected by generic personal and environmental factors, which can function as either barrier or facilitator. However, the presence of either transplant-related or generic barriers or facilitators of fruit and vegetable consumption are unexplored in RTR. Only two studies examined the perspectives and barriers of healthy dietary behaviors after renal transplantation [13,14]. The presence of both transplant-related and generic barriers and facilitators has been shown for physical activity in chronic kidney disease (CKD) and has important consequences for its management [15]. While generic barriers can be addressed by public health measures or by general healthcare professionals, transplant-related barriers may require specific measures from healthcare professionals involved in renal transplant care. Hence, for targeted nutritional counseling post-transplantation more in-depth knowledge of the barriers and facilitators is required.

Social support is a key facilitator in changing dietary habits and associated with dietary adherence in patients with ESRD [16,17]. Partners and family members are important supportive resources, especially when involved in the daily food preparation. Moreover, healthcare professionals also have an important role in addressing dietary behavior, for nutritional counseling is integrated in the routine care after renal transplantation. Thus, it is important to include the experiences of family members and healthcare professionals in addition to those of RTR themselves. The aim of this focus group study was to explore the barriers and facilitators of fruit and vegetable consumption in RTR, their family members and healthcare professionals. Furthermore, it was also explored if certain barriers and facilitators were related to the transplant-setting. This way, targeted support strategies can be developed to improve nutritional counseling and facilitate RTR with incorporation of the dietary recommendations into their daily lives.

Box 1Outline of Standard Nutrition Care.The standard nutritional counseling after renal
transplantation consists of at least one inpatient visit by a renal dietician
during the hospital admission (7–10 days), followed by one outpatient visit.
During the inpatient visits, the dietician focus on several dietary measures,
e.g., adequate nutritional intake to support post-operative recovery, the
avoidance of high-risk infectious food products and, if indicated, cessation
of pre-existent dietary restrictions. All renal transplant recipients (RTR)
receive a standard brochure with all dietary recommendations. The outpatient
visit is individualized according to the needs and medical background of the
patient. For example, when an increased 24-h urinary sodium excretion is
observed, salt consumption is discussed. In case of excessive weight gain or
diabetes, attention is paid to the caloric intake, intake of
mono-/disaccharides and diet quality. If indicated, additional outpatient
visits or telephonic follow-up will be scheduled.

## 2. Materials and Methods

### 2.1. Design and Participant Selection

This qualitative study was designed as a series of focus group discussions (FGDs) with RTR, their family members and healthcare professionals involved in renal transplant care. RTR and their family members were invited to participate when 18 years and older, transplanted within the last five years, with a stable preserved renal function (estimated glomerular filtration rate (eGFR) > 20 mL/min; of the recipient) and sufficient command of the Dutch language. A purposive sampling strategy was used to select eligible participants. Recruitment took place in collaboration with the nephrologists at the outpatient clinic of the Nephrology Department of the University Medical Center Groningen (UMCG). We aimed to recruit RTR and family members with different sociodemographic and medical backgrounds. Furthermore, different healthcare professionals were invited, including nephrologists, nurse practitioners, dieticians and social workers. All participants received information about the content of the study, emphasizing confidentiality and anonymity. This study was approved by the Institutional Review Board of the UMCG (METc 2017/482) and written informed consent was obtained from all participants.

### 2.2. Interview and Data Collection

Seven focus groups (three with RTR, three with family members and one with healthcare professionals) were organized between November 2017 and April 2018 with four to seven participants in each group, until data saturation occurred (Table 1). Data saturation was reached when no new information was obtained from the FGDs and this was assessed by comparing the codes and identified themes in consecutive FGDs [18]. The FGDs were organized in a neutral meeting room at the hospital and the duration was approximately two hours per session. The group discussion was moderated by one of the investigators (K.B.-M.) and audio recorded. One investigator (O.P.) was an observer during the sessions and took field notes about the nonverbal reactions and group dynamics. Both investigators were trained in performing FGDs and were not involved in the treatment of the participants. The interview was semi-structured with a question route that was designed by Krueger [19] and consisted of five steps: (1) introduction of participants, (2) introductory question, (3) transition question to bring the discussion towards the key points, (4) key questions and (5) an ending question with the possibility to add information that has not been discussed yet. The questions were open-ended and responses were explored by using probe questions. Participants were asked about the impact of the renal transplantation, medications and prior dialysis treatment (if applicable) on the dietary habits, as previous studies suggested that these factors affect dietary habits in RTR [7,11,12,13]. Participants were also asked about other barriers and facilitators of fruit and vegetable consumption (Appendix A). After the group discussion, participants filled out a brief questionnaire for collection of sociodemographic information and food habits. Finally, after completion of the FGD, by way of cognitive debriefing, the participants received a brief explanation of the background and goals of this study.

### 2.3. Conceptual Framework

The discussion guide was developed in line with the social cognitive theory (SCT), a framework for people’s behavioral choices and maintenance of health behavior in which personal, environmental and behavioral factors interact [20]. This model was selected because it acknowledges both individual and environmental determinants of health behavior, including dietary behaviors such as fruit and vegetable consumption [21]. Key elements include knowledge of health risk and behavior, self-efficacy, outcome expectations, individual health goals, social support and other barriers and facilitators of health behavior [20]. Considering the implementation in clinical practice, the main goal was to identify barriers and facilitators. Therefore, all identified themes at individual and environmental level were classified as either barrier or facilitator.

### 2.4. Data Analysis

The FGDs were fully transcribed and reviewed line-by-line by one of the investigators (K.B.-M.). Transcripts were analyzed by using content analysis. Two investigators (K.B.-M. and O.P.) coded the transcripts of three sessions independently. A codebook was made subsequently and used to code the complete transcript, with the ability to add new codes (K.B.-M. and O.P.). Inconsistencies were reviewed and discussed until consensus was reached, with assistance of a third reviewer (M.C.E.B.-F.). The participants received a summary with the key points of the discussion and they were invited to give feedback to ensure the findings were in line with the participants’ perspectives. The codes were grouped in themes and the identified themes were categorized in personal and environmental factors, as either barrier or facilitator. Finally, quotations were selected to illustrative the themes that derived from the analyses. Quotations were translated from Dutch to English (K.B.-M. and G.K.P.). The qualitative data program “ATLAS.ti” was used for the data analysis (ATLAS.ti, version 8.3.2, Atlas.ti Scientific Software Development GmbH, Berlin, Germany).

## 3. Results

### 3.1. Baseline Characteristics

A total of 19 patients, 15 family members and five healthcare professionals participated in seven FGDs. Four RTR and four family members also gave informed consent for participation, but were unable to attend one of the FGDs due to an intercurrent illness (*n* = 3), unavailability at the proposed dates and timeframes (*n* = 4) or unknown reason (*n* = 1). Six family members were related to patients that also participated in this study. In the FGD with healthcare professionals participated one nephrologist, two dieticians, one nurse practitioner and one social worker. The mean age of patients was 58 ± 11.8 (standard deviation (SD)) years, of family members 65 ± 7.2 SD years and of healthcare professionals 46 ± 14.8 SD years. On average, the healthcare professionals were 10 years (range 4–26 years) involved in renal transplant care. A sufficient daily vegetable consumption was reported by 26 percent of the patients and 13 percent of the family members. All baseline characteristics of the patients and family members are shown in Table 2.

### 3.2. Barriers and Facilitators

Several barriers and facilitators were identified from the FGDs and categorized in personal or environmental factors (Table 3). All factors that were related to the transplantation were categorized separately for their role as either barrier or facilitator. These transplant-related factors were further divided in ‘transition in diet’, ‘medication’, ‘physical health’ and ‘priorities after transplantation’. Generic personal factors included ‘food literacy levels’, ‘attitudes and motivation’, ‘self-efficacy’, ‘financial resources’ and ‘pre-existent food habits and preferences’. Environmental factors encompassed ‘social support’ and ‘the role of the partner as food gatekeeper’. While some barriers and facilitators were mentioned for fruit and vegetable consumption specifically, other factors were also mentioned in the context of other dietary measures.

#### 3.2.1. Transplant-Related Barriers and Facilitators

*Transition in diet*. Several barriers of fruit and vegetable consumption were related to the transition in diet after renal transplantation due to the cessation of pre-existent dietary restrictions. In some cases, there was insecurity or fear to eat potassium-rich foods again, such as fruit and vegetables, and this was also recognized by healthcare professionals.

“*It was kind of a change after transplantation, which was a surprise for my wife. She was allowed to eat healthy again. That doesn’t damage my kidneys anymore? It was difficult to have confidence that you could eat healthy again*.”
M, 63 years, partner.

Other participants indicated difficulties to incorporate new habits, such as regularly eating fruit, into their daily lives.

“*But I do have problems with eating fruit; at a certain point before transplantation I was not allowed to eat a lot of fruit. Thus, that really was a punishment for me. But since I am allowed to eat it again, I just forget it quite often*.”F, 56 years, patient.

It was also noted that some participants were still practicing the old dietary regime and continued the potassium restriction. In this case, participants assumed that this diet was still necessary or healthier for their kidney.

“*Yes, huge salt restriction, protein restriction, potassium restriction. And I can say now, a world has re-opened up for me. But if you’re following a diet for 10 years and you stick to it, it is always a good choice*.”M, 67 years, patient.

Participants relied heavily on their laboratory tests (e.g., renal function) in changing dietary behavior; it was assumed that stable blood results reflected a good practice of dietary measures. Additionally, it was observed that most participants mainly focused on the current dietary restrictions (e.g., avoidance of excessive salt and high-risk infectious food products). These restrictions dominated their food choices at home and at social eating occasions.

“*No, he says, I can’t have that. And no, then it won’t be eaten. Then he has the book and I say, now read in the book. Well, with that book in his hands he says, this is not allowed and that is not allowed. Yes, he is very strict in dieting*.”F, 70 years, partner.

*Medication*. The cravings and insatiable hunger that occurred as side effects of prednisolone also influenced food choices. However, these side effects diminished over time with tapering of the prednisolone dose. Additionally, participants avoided some fruits (e.g., grapefruits) because of the interaction with immunosuppressive medications.

“*What is a real punishment is that grapefruit is not allowed. It is my favorite fruit. I asked it again last time, maybe once in the three months or six months. No*.”F, 56 years, patient.

*Physical health*. While most patients indicated an improvement in physical functioning, some mentioned that coping with fatigue was still an issue. This negatively influenced food choices, as it was more likely to choose unhealthy convenience foods.

“*Fatigue I think. If I feel tired, I grab something with sugar more easily. Of course, I am just as tired now. Compared with a healthy person, although they also feel tired sometimes. And then you grab food with fast energy. Yes, you don’t feel like peeling a tangerine*.”F, 32 years, patient.

Recovery of uremic symptoms improved appetite and made it easier to follow the dietary recommendations.

“*Yes, and then suddenly you’re not nauseas anymore, and you think hallelujah. Really, then you can just eat again*.”F, 32 years, patient.

Following dietary measures for diabetes or for weight loss influenced food choices in a positive way; for example, refined carbohydrate sources, such as pasta or potatoes, were replaced by healthier alternatives and this was often accompanied by the consumption of more vegetables.

“*Vegetables are not a problem. Especially now we changed our lifestyle. So, less carbohydrates and that mean just a lot of vegetables*.”F, 56 years, patient.

*Priorities after transplantation*. Most patients mentioned a sense of responsibility for their own health and this resulted in a strong motivation to take care of their kidney. This was driven by fear of graft loss as well as a sense of gratitude towards the donor. A healthy diet was regarded as one of the measures that supports protection of the graft by some participants.

“*And indeed, my husband says, ‘I have a good kidney now, so I want good blood sugar levels and a good blood pressure.’ So yes, we take those things into account*.”F, 52 years, partner.

For other participants diet was not a main priority after transplantation. Some patients mentioned a relief after the release of dietary restrictions and they decided to eat whatever they liked without regard for the dietary advices.

“*Yes, and after the transplantation, I gained a lot of weight, about 20kg. Well, that was it, that eating, I went crazy. I felt so relieved at that time, I could not deal with it for a while*.”F, 32 years, patient.

For others, disease management was dominated by other factors, such as (infectious) complications, frequent hospital visits and complex medication regimes. Additionally, in some cases, the engagement in social participation activities (e.g., work, recreational activities with family) was prioritized.

“*Well, you need to come here quite often for regular checks. And then you are also invited to visit a dietician, an extra visit. I really wanted to go back to work. That kept me very busy. I also wanted to spend more time with my children. So, another dietician visit was just not my priority*”F, 47 years, patient.

From the healthcare professionals’ perspective, other aspects of routine care after transplantation were regarded more relevant for (graft) survival, e.g., medication compliance, medical treatment of hypertension and diabetes and managing complications. The healthcare professionals also experienced time constraint during the consultation as a barrier in discussing dietary behaviors.

“*Well, yes, there are a lot of things to do. So, yes, those “wins”, are not necessarily quick wins, but at least those things that are the most important determinants of the survival after transplantation. And yes, nutrition is a small part, but there are many other factors that are more important in my opinion*.”M, 36 years, nephrologist.

#### 3.2.2. Personal Barriers and Facilitators

*Food literacy*. Knowledge of food and cooking skills were often mentioned as facilitators of fruit and vegetable consumption and adherence to dietary recommendations. A variety of skills were mentioned, such as the ability to read food labels and to modify standard recipes. For example, in order to avoid excessive salt consumption, salt was substituted by herbs, spices or vegetables to flavor meals.

“*My son cooks and uses more vegetables. Previously, I found meat more important than vegetables. But now, it is the other way around, vegetables are more important. Food is very tasty with the use of onions and red pepper, that kind of things. That really flavors your food*.”F, 62 years, mother.

These elements are part of the broader concept food literacy. This refers to the capability to make healthier food choices in different contexts and encompasses the knowledge, skills and behavior to plan, manage, select, prepare and eat food healthfully [22]. Few participants indicated difficulties with dietary measures after transplantation, for example with seasoning of food without salt.

“*Well, yes, salt, you shouldn’t eat that too much. So I mean without salt or sodium or whatever. But then they say, sodium is salt, and that is also not good for you. But what do you have to do if normal salt is not an option*?”F, 70 years, partner.

The importance of the ability to bring dietary measures into practice was also noted by dieticians. However, it was acknowledged that teaching patients skills was not part of regular counseling yet.

“*Sometimes I doubt if we should do more with real, uh, the part of food skills. At the moment it is mainly transfer of information or advices. While some patients say, yes I understand this. But understanding and then translating this to actual practice*.”F, 25 years, dietician.

For fruit and vegetable consumption specifically, it was noted that many participants overestimated their daily intake of fruit and vegetables.

*Attitude and motivation*. Fruit and vegetable consumption was also influenced by either a positive or negative attitude of the participant towards the health benefit of fruit and vegetables. When these health benefits were not acknowledged, this resulted in less motivation. Lack of time was also mentioned as barrier for healthy food choices. The motivations that were given for eating healthily were divided in: pleasure in cooking, enjoying eating healthy food and improvement of wellbeing.

“*I feel a lot more energetic. And now you also are forced to pay attention to your food. And I think I feel better with it, yes*.”F, 49 years, patient.

*Other.* Participants with high self-efficacy found it easier to practice the dietary measures. Pre-existent food habits and routine, taste preference and financial resources also influenced fruit and vegetable consumption in either a positive or negative way.

“*I do eat a tangerines sometimes, but other than that I hardly eat any fruit. I am just not really a fruit-type of person*.”F, 70 years, partner.

“*What pops up in my mind is the financial part. People find it difficult to eat healthy, because it is financially difficult. They do not have the opportunity to visit the market. They know, but they can’t find ways to get there. Those are the kind of problems I come across*.”F, 51 years, social worker.

#### 3.2.3. Environmental Barriers and Facilitators

Social support. Social support in following dietary measures was highly valued by the participants. Most family members participated in the diet of the RTR. This facilitated the ease of cooking and was also seen as part of solidarity. Additionally, some family members acknowledged a shared health benefit.

“*You adapt yourself, in terms of your partners’ eating habits. What is allowed and what is not. You accommodate to it. It sounds like a cliché, but you support each other for better and for worse. You want to stay together as long as possible to enjoy each other’s company. You would do anything for that*.”M, 68 years, partner.

Many family members were the “food gatekeepers” and responsible for grocery shopping and preparing meals. Some were actively involved in nutritional counseling, which facilitated the practice of dietary measures.

“*I never look at recipes, but as I say, I have a good cook. I have nothing to complain, I have a good cook and that is true. She stays on top of it*.”M, 67 years, patient.

Others mentioned difficulties with the dietary advices and expressed feelings of helplessness or frustrations.

“*And then when I’m cooking and he says: you’re using way too much salt, that isn’t allowed! Well, then I don’t know anymore. Those are the kinds of problems I face. … But what am I supposed to do, I have to deal with it on my own. And then I think, what am I supposed to I cook then*!”F, 70 years, partner.

Family members found the involvement in nutritional counseling beneficial for their understanding of the dietary measures and it also supported the recall of advices at home.

## 4. Discussion

In this focus group study, we systematically identified transplant-related and generic barriers and facilitators of fruit and vegetable consumption in RTR, their family members and healthcare professionals. RTR face unique challenges in modifying their dietary behavior after renal transplantation that entail transplant-related factors like the transition in diet, changes in physical health, medications and competing priorities. The generic factors found in this study mainly involved food literacy and social support. These findings underscore the need for additional targeted nutritional counseling strategies in the routine care of RTR.

Several transplant-related barriers were related to the transition in diet. This transition can be accompanied by a fear of complications or incorrect assumptions to continue the old dietary restrictions. A previous study also found that the fear of graft rejection leads to the avoidance of certain food products, but details about the type of foods were not given [23]. Moreover, we found that difficulties can arise with the incorporation of new dietary measures that are opposite to the ingrained food habits at time of ESRD. This difficulty with changing dietary habits was also previously identified for fluid intake in RTR [14]. We showed that these dietary transition barriers affect consumption of potassium-rich foods, including fruit and vegetables. Measurement of 24-h urinary potassium excretion could be a useful tool to identify RTR with persistent low potassium intake that require additional counseling [24].

It was also noted that food choices post-transplantation were still driven by dietary restrictions. This mainly involved the avoidance of excessive salt; this advice remains after transplantation. CKD patients traditionally receive nutrient-based recommendations, regarding salt, potassium, protein and phosphate. However, the importance of diet quality and whole dietary patterns is increasingly recognized for the health outcomes of general and CKD populations, including RTR [6,25,26]. In line with the general food-based nutritional guidelines, moving towards food-based recommendations could be a valuable alternative for RTR; it offers easier interpretable advice with more emphasis on diet quality and food products that positively influences health, such as fruit and vegetables.

In previous literature, only two qualitative studies have explored the barriers of RTR in adhering to a healthy diet [13,14]. One study also mentioned transplant-related barriers of eating healthily, namely the delight of eating without restrictions and side effects of prednisolone [13]. In the present study, the joy of eating without limitations was also found as a barrier for adherence to dietary recommendations. Additionally, we also identified the side effects of prednisolone as a barrier, but participants mentioned that these side effects diminished with tapering of the dose to five milligrams/day. Furthermore, we found competing priorities to be a barrier, such as social participating activities or preoccupation with disease management. Acknowledging the presence of other priorities is essential for optimal delivery of dietary advice for RTR.

The sense of responsibility for the new kidney was identified as a transplant-specific facilitator of good dietary practice. This motivated RTR to adhere to supportive measures, such as the dietary recommendations. This preoccupation for self-care activities to support overall health and protect the kidney was also found in a previous qualitative study [27]. Nevertheless, in most of these motivated participants fruit and vegetable consumption did not meet the daily recommendations. This highlights that the removal of barriers may be more important to change their dietary behavior than increasing their intrinsic motivation.

Among the generic personal barriers and facilitators, several elements of food literacy were identified as facilitators of fruit and vegetable consumption and adherence to dietary recommendations. Adequate food literacy is important to understand nutritional information and the ability to translate this knowledge into practice. Higher food literacy levels are associated with more fruit, vegetable and fish consumption in healthy individuals [28]. Food literacy is derived from the broader concept of health literacy, which involves the capabilities to access, understand, appraise and apply health-related information [29]. The importance of health literacy is increasingly recognized for its impact on health outcomes [30], leading to development of health literacy interventions [31]. Similarly, enhancing food literacy levels by educational sessions could support RTR to improve their dietary habits. However, more knowledge is needed about the food literacy levels of RTR, its relationship with diet quality and the benefit of enhancing these capabilities in the context of dietary behavior and health outcomes.

On an environmental level, the lack of social support was a generic barrier of practicing dietary measures. This was identified previously as a barrier of following a low-salt, low-cholesterol diet after renal transplantation [14]. While partners were often the “food gatekeepers”, some were not involved in nutritional counseling and struggled with dietary advices. This could negatively influence dietary adherence and underscores the need for active involvement of the partner in dietary education.

Even though this study provides important insights into the barriers and facilitators of fruit and vegetable consumption, there were several limitations. One of the limitations is the risk of bias caused by socially desirable responses in line with group norms that occur in FGDs [32]. However, the advantage of FGDs is the comprehensive approach to capture experiences and opinions and the group interaction that supports exploration of individual and shared perspectives [32]. Moreover, although we were able to include participants with different sociodemographic and medical backgrounds, only Dutch-speaking participants were included to avoid miscommunication during the sessions. This limits generalizability to populations with different cultural backgrounds, where other barriers may exist. Additionally, we cannot rule out the occurrence of selection bias, as participants that prioritize lifestyle measures may have been more likely to participate in this study. We only included participants from one transplantation center and were not able to compare the identified themes in RTR that receive the nutritional counseling elsewhere. Finally, the relative contribution of each barrier and facilitator to the dietary behaviors needs to be verified in a larger RTR population as well as RTR populations in other sociodemographic contexts.

Several implications can be derived from our findings. First, whereas this study focused on fruit and vegetable consumption, it must be noted that these findings should be placed in the broader perspective of lifestyle behaviors after renal transplantation, and that the complexity and interrelatedness of lifestyle factors warrants an integrated approach, rather than focus on single issues. For example, post-transplant weight gain was not only related to poor diet quality, but mainly due to a lack of physical activity, whereas caloric intake was irrelevant. Furthermore, while generic barriers could be addressed by public health measures or general healthcare professionals, transplant-related barriers require specific measures that are preferably incorporated in the routine care of RTR. The timing of dietary counseling should be tailored to the patient’s personal journey and needs. Moreover, specific attention for dietary transition barriers is required in RTR, especially in those who had pre-existent dietary restrictions. The importance of consuming food products with a positive health influence, instead of single nutrients, should also be emphasized. Finally, it highlights the potential benefit of enhancing food literacy levels and the need for active involvement of family members. Given the complex interplay of different lifestyle factors, the dietary counseling should be part of integrated lifestyle management, in which diet, physical activity and barriers of behavioral changes are addressed in the context of the individual patient and the disease history.

## 5. Conclusions

In conclusion, this study provides an in-depth exploration of transplant-related and generic barriers and facilitators for fruit and vegetable consumption in RTR. These findings can be used for the development of targeted nutritional counseling strategies in renal transplant care.

## Figures and Tables

**Table 1 nutrients-11-02427-t001:** Characteristics of focus groups of RTR, family members and health-care professionals.

Focus Group	Number of Participants	Age Range	Date of FDG	Gender		Time since Tx in Months	Dialysis before Tx	Potassium-Restricted Diet before Tx
				M	F			
RTR	6	40–73	November 2017	3	3	4–57	4	3
RTR	6	32–66	December 2017	3	3	8–55	4	2
RTR	7	46–68	January 2018	6	1	4–57	3	3
Family members	5	49–68	February 2018	3	2	7–60 *	2 *	1 *
Family members	4	52–73	March 2018	0	4	5–39 *	3 *	2 *
Family members	6	62–77	April 2018	3	3	3–26 *	3 *	4 *
Healthcare professionals	5	25–61	April 2018	1	4			

* Of renal transplant recipient. Abbreviations: F, female; FDG, focus group discussion; M, male; RTR, renal transplant recipients; Tx, transplantation.

**Table 2 nutrients-11-02427-t002:** Participants characteristics of RTR (*N* = 19) and family members (*N* = 15).

Characteristics	RTR	Family Members
Demographics		
Age (mean, SD)	58 ± 11.8	65 ± 7.2
Gender (*N*, % male)	12 (63)	6 (40)
Highest level of education (*N*, %)		
Primary education		
Secondary education	5 (26)	6 (40)
Vocational education	12 (63)	2 (13)
Tertiary education (college/university)	2 (11)	6 (40)
Missing		1 (7)
Work status (*N*, %)		
Full-time	3 (16)	3 (20)
Part-time	3 (16)	2 (13)
Retired	6 (31)	9 (60)
Disabled due to health	4 (21)	
Unemployment	3 (16)	
Missing		1 (7)
Medical background		
Primary renal disease (*N*, %)		
Primary glomerular disease	7 (37)	
Tubulointerstitial disease	2 (11)	
Systemic disease	4 (21)	
Hereditary disease	6 (31)	
Time since Tx in months (mean, SD)	24 ± 20.5	19 ± 17.4 *
Dialysis before transplantation (*N*, %)	11 (58)	8 (53) *
Dialysis duration in months (mean, SD)	30 ± 23	25 ± 17.6 *
eGFR (mL/min * 1.73 m2)	50.8 ± 11	
Plasma potassium (mmol/L)	4.2 ± 0.4	
BMI (mean, SD)	29 ± 6.4	26 ± 4.6
Hypertension (*N*, %)	13 (68)	5 (33)
Diabetes Mellitus (*N*, %)	1 (5)	1 (7)
PTDM (*N*, %)	3 (16)	
Food habits		
Potassium restriction prior Tx (*N*, %)	8 (42.1)	7 (46.6) *
Vegetable consumption > 200 g/day (*N*, %)	5 (26)	2 (13)
Salt consumptions g/day (mean, SD)	8.5 ± 3.7	

* of renal transplant recipient. Abbreviations: BMI, body mass index; eGFR, estimated glomerular filtration rate; g/day, grams per day; PTDM, post-transplantation diabetes mellitus; *N*, number; RTR; renal transplant recipients; SD, standard deviation; Tx, transplantation.

**Table 3 nutrients-11-02427-t003:** Overview of transplant-related, personal and environmental barriers and facilitators of fruit and vegetable consumption in RTR.

Theme	Barrier	Facilitator
Transplant-related factors		
Transition in diet	Holding on to restricted diet	Freedom of choice in fruit/vegetables
	Struggle with new routine	
	Insecurity/fear	
	Focus on dietary restrictions *	
Medication	Cravings/insatiable hunger prednisolone *	
	Food interaction medication	
Physical health	Fatigue/lack of energy *	Recovery uremic symptoms *
		Dietary measures diabetes
Priorities/Goals after Tx	Burden of disease management *	Protecting the transplant *
	Social participation activities *	
	Enjoying life *	
	Not main priority nephrologist *	
Generic factors		
Personal		
Food literacy	Limited food literacy *	Adequate food literacy *
	Overestimation of vegetable consumption	Ability to practice dietary measures *
	Difficulties with practicing diet *	
Attitudes/Motivations	Negative attitude/Lack of motivation:	Positive attitude/Motivation:
	No perceived health benefit	Perceived health benefit
	Too much effort/lack of time	Pleasure/fun in cooking *
	Recommended amount too much	Enjoying healthy food *
		Improvement well-being *
Other	Lack of routine/poor pre-existent habits	Routine/pre-existent habits
	Food/taste preference	Food/taste preference
	Limited financial resources *	Self-efficacy *
Environmental		
Social environment	Lack of social support *	Social support *
	Partner is food gatekeeper *	Partner is food gatekeeper *

* These barriers and facilitators were also mentioned in the context of other dietary measures. Abbreviations: RTR, renal transplant recipients; Tx, transplantation.

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
