# Peer review of "Barriers and Facilitators of Fruit and Vegetable Consumption in Renal Transplant Recipients, Family Members and Healthcare Professionals—A Focus Group Study"

_nutrients, 2019, doi:10.3390/nu11102427_

Round 1
Reviewer 1 Report
Dear Authors,
Although, the current research study is qualitative one, it addresses very important question about the barriers and facilitators in general, of poor fruit and vegetables consumption after renal transplantation, and if certain barriers and facilitators are more transplant-related. Conclusions made by the authors are in congruent with the research objectives, however, methodology lacks some pertinent details that deserves authors attention such as -
Please specify, if any training was conducted/imparted to the interviewer(s) for conducting this FGDs? How questionnaire was developed for this semi-structured interview? Was it based of some literature review or expert opinions? There is inconsistency in gender distribution and number of participants in each focus group. Was there any specific reason for this inconsistency since your sample selection was purposive? Please specify which health care professionals (MD, nurses, etc.) with their professional credentials participated in FGD? and how long are they involved in caring for RTR? How data saturation was determined, when there was only one focus group with 5 health care professionals? Was there any cognitive debriefing provided to participants immediately after the FGDs?
Reviewer 2 Report
The main problem connected with diet after transplantation is an excessive weight gain. One of the patient told during conversation "I gained a lot of weight, about 20 kg". The issue of balance fat and carbohydrate consumption with the energy expenditure is - in my view - even more important than insufficient fruit and vegetable intake. Obesity becomes crucial driver of diabetes post transplantation. These matters should be discussed with patients or at least mentioned in the discussion of the results.
Reviewer 3 Report
This manuscript is worth to publish with the current form.
I have nothing to comment or edit.
